# Transcriptomic Analysis Reveals Early Alterations Associated with Intrinsic Resistance to Targeted Therapy in Lung Adenocarcinoma Cell Lines

**DOI:** 10.3390/cancers16132490

**Published:** 2024-07-08

**Authors:** Mario Perez-Medina, Jose S. Lopez-Gonzalez, Jesus J. Benito-Lopez, Santiago Ávila-Ríos, Maribel Soto-Nava, Margarita Matias-Florentino, Alfonso Méndez-Tenorio, Miriam Galicia-Velasco, Rodolfo Chavez-Dominguez, Sergio E. Meza-Toledo, Dolores Aguilar-Cazares

**Affiliations:** 1Laboratorio de Cancer Pulmonar, Departamento de Enfermedades Cronico-Degenerativas, Instituto Nacional de Enfermedades Respiratorias Ismael Cosio Villegas, Ciudad de Mexico 14080, Mexico; mperezm1518@alumno.ipn.mx (M.P.-M.); sullivan.lopez@iner.gob.mx (J.S.L.-G.); jareb.benito.l@comunidad.unam.mx (J.J.B.-L.); miriam.galicia@iner.gob.mx (M.G.-V.); rodolfo_chvz@comunidad.unam.mx (R.C.-D.); 2Laboratorio de Quimioterapia Experimental, Escuela Nacional de Ciencias Biologicas, Instituto Politecnico Nacional, Ciudad de Mexico 14080, Mexico; smezat@ipn.mx; 3Posgrado en Ciencias Biologicas, Universidad Nacional Autonoma de Mexico, Ciudad de Mexico 14080, Mexico; 4Centro de Investigacion en Enfermedades Infecciosas, Instituto Nacional de Enfermedades Respiratorias Ismael Cosío Villegas, Ciudad de Mexico 14080, Mexico; santiago.avila@cieni.org.mx (S.Á.-R.); maribel.soto@cieni.org.mx (M.S.-N.); margarita.matias@cieni.org.mx (M.M.-F.); 5Laboratorio de Biotecnologia y Bioinformatica Genomica, Departamento de Bioquimica, Escuela Nacional de Ciencias Biologicas, Instituto Politecnico Nacional, Ciudad de Mexico 14080, Mexico; amendezt@ipn.mx

**Keywords:** lung adenocarcinoma, drug-tolerant persister (DTP) cells, intrinsic resistance, tyrosine kinase inhibitors (TKIs), erlotinib, osimertinib, senescent cells, CD74

## Abstract

**Simple Summary:**

Lung adenocarcinoma, the most prevalent type of lung cancer, presents a significant treatment challenge owing to drug resistance. This study aimed to investigate the role of long non-coding RNAs (lncRNAs) in promoting intrinsic resistance in three lung adenocarcinoma cell lines from the onset of treatment. Drug-tolerant persister (DTP) cells, a subset of cancer cells with the ability to survive and proliferate after exposure to therapeutic drugs, were generated. RNA sequencing was used to investigate the differential expression of lncRNAs, and the clinical relevance of lncRNAs was assessed in a cohort of lung adenocarcinoma patients from The Cancer Genome Atlas. Knockdown of these lncRNAs increases the sensitivity of the analyzed cell lines to therapeutic drugs. This study provides an opportunity to investigate the role of lncRNAs in the genetic and epigenetic mechanisms underlying intrinsic resistance.

**Abstract:**

Lung adenocarcinoma is the most prevalent form of lung cancer, and drug resistance poses a significant obstacle in its treatment. This study aimed to investigate the overexpression of long non-coding RNAs (lncRNAs) as a mechanism that promotes intrinsic resistance in tumor cells from the onset of treatment. Drug-tolerant persister (DTP) cells are a subset of cancer cells that survive and proliferate after exposure to therapeutic drugs, making them an essential object of study in cancer treatment. The molecular mechanisms underlying DTP cell survival are not fully understood; however, long non-coding RNAs (lncRNAs) have been proposed to play a crucial role. DTP cells from lung adenocarcinoma cell lines were obtained after single exposure to tyrosine kinase inhibitors (TKIs; erlotinib or osimertinib). After establishing DTP cells, RNA sequencing was performed to investigate the differential expression of the lncRNAs. Some lncRNAs and one mRNA were overexpressed in DTP cells. The clinical relevance of lncRNAs was evaluated in a cohort of patients with lung adenocarcinoma from The Cancer Genome Atlas (TCGA). RT–qPCR validated the overexpression of lncRNAs and mRNA in the residual DTP cells and LUAD biopsies. Knockdown of these lncRNAs increases the sensitivity of DTP cells to therapeutic drugs. This study provides an opportunity to investigate the involvement of lncRNAs in the genetic and epigenetic mechanisms that underlie intrinsic resistance. The identified lncRNAs and CD74 mRNA may serve as potential prognostic markers or therapeutic targets to improve the overall survival (OS) of patients with lung cancer.

## 1. Introduction

According to GLOBOCAN, lung cancer is the leading cause of death in both developed and developing countries [1]. Lung cancer is classified into small-cell lung carcinoma (SCLC) and non-small-cell lung carcinoma (NSCLC) [2]. NSCLC is the most frequent type, comprising three major histological types: large-cell carcinoma, squamous cell carcinoma, and adenocarcinoma (the most common histologic type) [3]. The identification of genetic alterations has led to the development of targeted therapies for treating early and advanced lung cancer, resulting in higher overall survival of patients [4]. Based on the clinical guidelines outlined by the National Comprehensive Cancer Network (NCCN) for the treatment of lung cancer, patients with tumors harboring activating mutations in the epidermal growth factor receptor (*EGFR*) gene require therapy with tyrosine kinase inhibitors (TKIs) [4,5,6]. Depending on the *EGFR* mutation status, patients were initially treated with EGFR-specific TKI, which interact reversibly with the ATP-binding site of EGFR, for example, erlotinib and gefitinib (first-generation EGFR TKIs). Currently, the second- and third-generation TKIs afatinib and osimertinib, which irreversibly bind to the kinase domain of the EGFR, have become the leading choice of treatment for lung cancer harboring *EGFR* mutations [7,8,9]. Although most patients initially respond favorably to TKIs, approximately 16–30% of treated patients do not show clinical benefit, mainly due to the development of drug resistance, which limits the initial efficacy of these targeted therapies [10,11,12].

The biological causes of resistance, particularly those associated with acquired resistance, have been investigated extensively. Tumors comprise distinct cell populations that show genetic and phenotypic heterogeneity, and populations with lower susceptibility to chemotherapy and targeted therapies have been reported [13]. Upon exposure to therapeutic compounds, these drug-tolerant persister (DTP) cells survive by exhibiting alterations, such as entering a state of dormancy, showing decreased proliferation and changes in oxidative and lipid metabolism, chromatin remodeling, increased drug metabolism, alterations in drug efflux systems, and cell-cycle arrest [14,15]. In addition, DTP cells can generate clones with characteristics such as a mesenchymal–epithelial transition (MET) phenotype, increased migration, and promotion of metastatic activity [16].

Intrinsic resistance, as an additional phenomenon, has recently been considered a significant factor in tumor recurrence [17]. In this phenomenon, populations of tumor cells display alterations before drug exposure or immediately after the initial cycles of treatment that allow them to survive the treatment, eventually giving rise to tumors with reduced sensitivity to therapeutic drugs [18]. However, the molecular mechanisms underlying intrinsic resistance remain poorly understood [19]. Variability among the transcriptomic profiles of the tumor cell population has been proposed as the leading cause of intrinsic resistance [20]. In vitro studies in lung cancer, osteosarcoma, and melanoma cell lines have reported that a fraction of tumor cells survive after initial exposure to therapeutic drugs and that residual cells are detected in a latent state [21,22,23]. In several types of cancer, it has been suggested that intrinsic resistance is related to non-genetic mechanisms [24]. Further studies are needed to better understand the molecular mechanisms underlying the emergence of DTP cells in *EGFR*-mutated lung adenocarcinoma and their role in treatment resistance. A possible mechanism for intrinsic resistance is immediate reprogramming of the transcriptome, mediated by the deregulation of long non-coding RNAs (lncRNAs), which allows tumor cells to modulate gene expression to resist the cytotoxic effects of various antitumor agents [25].

The present study aimed to identify early transcriptomic alterations associated with the DTP state in *EGFR*-mutated lung adenocarcinoma cell lines after a single exposure to TKIs. Our results demonstrated that cell lines with *EGFR* mutations show different degrees of susceptibility to TKIs and that cell-cycle arrest is mainly induced in the G0/G1 phase. After exposure to erlotinib or osimertinib, residual β-galactosidase-positive cells were detected, suggesting a senescence-like phenotype in the DTP cells, as previously reported [26]. RNA sequencing and differential gene expression analyses showed that DTP cells from the cell lines exposed to TKIs displayed specific overexpression of lncRNAs, suggesting that these lncRNAs may act as intrinsic resistance factors that may be deregulated after the onset of treatment. Using TCGA data, the detected lncRNAs were significantly associated with worse overall survival (OS) in lung adenocarcinoma patients. In addition, we found that osimertinib-induced DTP cells overexpressed the protein-coding gene *CD74*, and the corresponding protein was detected in biopsies from LUAD patients. Moreover, knockdown of these DTP-associated lncRNAs or *CD74* favored cell death upon TKI exposure. The expression of these lncRNAs and *CD74* was validated in lung adenocarcinoma biopsies. Therefore, the identified lncRNAs and *CD74* could be employed as novel predictive biomarkers or therapeutic targets to increase the effectiveness of targeted therapy and improve the clinical outcomes of patients with lung adenocarcinoma.

## 2. Materials and Methods

### 2.1. Cell Culture

The *EGFR*-mutant lung adenocarcinoma cell lines used in this study included HCC827 (harboring the E746-A750 activating mutation) and HCC4006 (harboring the L747-E749 + A750P activating mutation), both of which exhibit *EGFR* exon 19 deletions and are sensitive to erlotinib. The H1975 cell line was also incorporated, characterized by *EGFR* mutation (L858R/T790M) and resistance to erlotinib, but sensitivity to osimertinib. All cell lines were obtained from the American Type Culture Collection (ATCC; Manassas, VA, USA) and cells were cultured in RPMI-1640 medium (Sigma-Aldrich, St. Louis, MO, USA. Cat. No. R6504) supplemented with 10% fetal bovine serum (Hyclone, Issaquah, WA, USA, Cat. No. SH30070.03), and 1% antibiotics (complete medium). Cell cultures were maintained at 37 °C in a humidified incubator with 5% CO_2_.

### 2.2. Targeted Next-Generation Sequencing to Confirm Mutations in Lung Adenocarcinoma Cell Lines

To corroborate the mutational profiles of the cell lines obtained from ATCC and employed throughout the experiments, we conducted targeted sequencing using an Illumina TruSight Tumor 15 Kit (Illumina, San Diego, CA, USA, Cat. No. OP-101-1001). DNA extraction was performed using the Pure Link Genomic DNA Mini Kit (Cat. No. 1820-01). DNA concentration was assessed using Qubit (Invitrogen, Carlsbad, CA, USA). DNA libraries were prepared according to the manufacturer’s guidelines and sequenced using a MiSeq Illumina device. This panel was designed to identify clinically relevant mutations in the following 15 cancer driver genes: *AKT1*, *BRAF*, *EGFR*, *ERBB2*, *FOXL2*, *GNA11*, *GNAQ*, *KIT*, *KRAS*, *MET*, *NRAS*, *PDGFRA*, *PIK3CA*, *RET*, and *TP53*.

### 2.3. Therapeutic Compounds

Erlotinib-HCl (Cat. No. S1023) and osimertinib (Cat. No. S7297) were purchased from Selleckchem (Houston, TX, USA), dissolved in dimethyl sulfoxide (DMSO; Sigma-Aldrich, MO, USA), and diluted in complete media. The cell cultures were exposed to TKIs at concentrations of 25–200 nM. The range of concentrations of these compounds includes those reported for serum-treated patients [27,28]. Control cultures maintained in a complete medium or medium containing DMSO were used.

### 2.4. Dose–Response Curves to TKIs in EGFR-Mutant Lung Adenocarcinoma Cell Lines

Lung adenocarcinoma cell lines were cultured in T-25 flasks until they reached 80–90% confluence. The cells were then detached by treatment with trypsin, and the viability of the harvested cells always exceeded 95%. Cells were seeded in 96-well plates based on their growth rates: HCC827 and HCC4006 cells at 1 × 10^5^ cells/mL and H1975 cells at 1.5 × 10^5^ cells/mL. Cells were incubated for 24 h to allow their adhesion to plastic and to reach 60–70% confluence. The cells were then exposed to serial dilutions of TKIs and incubated for an additional 48 h. 

Microscopic observations were performed during drug exposure to assess the changes in cell density or cell death. These changes were documented using micrographs captured using an EVOS FL digital inverted microscope (Life Technologies, Carlsbad, CA, USA). To quantify the effect of treatment in lung adenocarcinoma cell lines, the MTT (3-(4,5-dimethylthiazol-2-yl)-2,5-diphenyltetrazolium bromide) assay was performed following the manufacturer’s instructions (Trevigen, Gaithersburg, MD, USA, Cat. No. 4890-25K). At the end of the drug exposure period, 10 μL MTT was added to each well. The plates were then incubated for 4 h in the dark at 37 °C. Formazan crystals were then dissolved in 150 μL of DMSO, and the absorbance was measured at 570 nm using a Multiskan Ascent plate reader (Thermo Fisher Scientific, Waltham, MA, USA). Cell viability was calculated relative to control, which was representative of 100% viability. Three independent experiments were performed, each one in triplicate.

### 2.5. Cell-Cycle Analysis

We considered it relevant to examine the effects of TKIs on cell-cycle phases. After TKI exposure, the adhered cells and those found in the supernatant were gently collected, combined, and washed with phosphate-buffered saline (PBS). Subsequently, the cells were fixed with cold 70% (*v*/*v*) ethanol and stored at −20 °C for a minimum of 24 h. After washing, cells were permeabilized with PBS containing 0.1% (*v*/*v*) Triton X-100 (Sigma-Aldrich, Cat. No. X100) and treated with RNase A (30 μg/mL) (Thermo Fisher Scientific, MA, USA, Cat. No. EN0531) to avoid nonspecific RNA staining. We performed DNA staining with 0.25 μg of propidium iodide (PI; Thermo Scientific, Lake County, IL, USA, Cat. No. 62248) for 30 min to acquire 25,000–30,000 events using a FACS Canto II flow cytometer. After establishing a Forward Scatter Area (FSC-A) vs. Forward Scatter Height (FSC-H) dot plot to discern individual cells (singlets), events were analyzed using an FSC-A vs. SSC-A dot plot. This population was analyzed using a PI-A vs. SSC-A dot plot. Finally, histograms were constructed to quantify the percentage of cells in each cell-cycle phase using the FlowJo v.10 software (FlowJo, Ashland, OR, USA). Two independent experiments were performed in triplicate to ensure the robustness and reproducibility of each assay. After treatment with TKIs, resistant cells were mainly arrested in the G0/G1 cell-cycle phase.

### 2.6. Detection of Beta-Galactosidase Positive DTP Cells

To explore whether the lung adenocarcinoma cell lines acquired a senescent-like phenotype in response to TKIs, we exposed the cell lines to osimertinib for 48 h in a four-chamber slide. Diluent- and complete media-treated cells were used as controls. After this time, the cell cultures were washed, and resistant cells were cultured in complete media for an additional five days. The cells were then washed with PBS, and beta-galactosidase staining was performed using the beta-galactosidase staining kit (Cell Signaling Technology, Danvers, MA, USA, Cat. No. 9860) following the manufacturer’s instructions. After treatment with a fixative solution for 10 min, cells were washed and incubated with beta-galactosidase staining solution (pH 6.0) for 4–5 h in a CO_2_-free incubator at 37 °C. Finally, cells were washed and mounted in 70% glycerol solution. Micrographs were acquired using an EVOS FL digital inverted microscope (Life Technologies). 

### 2.7. Quantification of Viable, Apoptotic, and Necrotic Cells

Microscopic observations of some cell cultures showed features of cell death, such as cell shrinkage, membrane blebbing, apoptotic bodies, and cell debris, regardless of the presence or absence of *EGFR* mutations. To corroborate these observations and quantify the percentages of viable, apoptotic, and necrotic cells, we used a FITC Annexin-V Kit from BD Pharmingen (Becton Dickinson, Franklin Lakes, NJ, USA, Cat. No. 556570). Briefly, cells were cultured in T-25 flasks under the same experimental conditions as previously described. After exposure to 100 nM (low concentration) or 200 nM (high concentration) of erlotinib or osimertinib, which induced the maximum effect according to the dose–response curves, floating dead cells were collected and adhered cells were harvested by trypsinization. The two cell fractions were mixed, washed with Ca^2+^ and Mg^2+^-free PBS, and rinsed with binding buffer. 3 × 10^5^ cells were stained with annexin-V/propidium iodide (PI), following the manufacturer’s instructions. After staining, 15,000 events were immediately acquired using a FACS Canto II flow cytometer (BD Biosciences). An FSC-A versus FSC-H dot plot was constructed to detect single cells in the gate. An FSC-A vs. SSC-A plot was then constructed, and the distribution of the collected events was analyzed using an annexin-V vs. propidium iodide dot plot. Finally, the proportions of viable, apoptotic, and necrotic cells were quantified using FlowJo v.10 software. Three independent experiments were performed in duplicates.

### 2.8. Transcriptome Sequencing

Transcriptome sequencing was performed in DTP cells to identify the transcriptional alterations related to intrinsic resistance to TKIs. After a single exposure to TKIs, we removed the floating dead cells after several washes with PBS and microscopically verified that only adherent cells remained in the culture. Adherent cells were harvested by trypsinization and showed viability of over 95%. Following the manufacturer’s instructions, total RNA was extracted from treated and untreated cells (control) using an RNAeasy Kit (Qiagen, Hilden, Germany). High-quality RNA samples (RIN > 8) were obtained using the TruSeq RNA Sample Prep Kit v2 (Illumina, San Diego, CA, USA). We sequenced the libraries employing the NextSeq 500 platform (Illumina, USA) at a depth of approximately 40 million reads, generating 75-bp paired-end reads. We sequenced at least two independent experiments in duplicates for each experimental condition.

### 2.9. Bioinformatics Analysis of DTP Cell RNA-Seq Data

Raw reads were subjected to quality control using FastQC (v.0.11.9) [29]. Trimmomatic (v.0.38) was used to filter reads with a Phred score > 26 [30], and the Cutadapt program (v.2.7) was used to remove adapter sequences as waste products [31]. The resulting clean reads were aligned with the GRCh38.95 reference genome using STAR (v.2.7.3a) [32], and gene abundance was estimated using RSEM (v.1.3.1) [33]. Poorly expressed genes were excluded, and only those with a mean count per million > 1 in all samples were considered. Differential gene expression (DGE) between DTP cells and untreated (diluent or media control) cells was assessed using the edgeR package (v.3.26.0) [34]. Significant gene expression was considered for |log2 fold| ≥ 1 and adjusted *p*-value of <0.05. Overexpressed genes were annotated using BioMart [35], and overexpressed lncRNAs were identified. For each TKI, the lncRNAs overexpressed in common by the DTP cells from the cell lines tested were identified using Venn diagrams.

### 2.10. Survival Analysis

The clinical relevance of the DTP-associated lncRNAs in the overall survival (OS) was analyzed in the LUAD cohort of TCGA using the GEPIA2 software [36]. Kaplan–Meier (KM) analyses and log-rank tests were performed. lncRNAs with a log-rank *p*-value < 0.05 were considered statistically significant. The expression levels of the clinically relevant DTP-associated lncRNAs found and CD74 were further analyzed in DTP cells.

### 2.11. RT–qPCR Validation

To validate the results obtained through bioinformatics analysis, total RNA from DTP cells and the corresponding controls was obtained using the MagMAX™ mirVana™ Kit (Thermo Fisher Scientific, MA, USA, Cat. No. A27828) following the manufacturer’s instructions. The concentration and purity of the RNA were analyzed using a NanoDrop spectrophotometer (Thermo Fisher Scientific). Purified RNA was reverse-transcribed into cDNA using SuperScript™ VI-LO™ (Thermo Fisher Scientific, Cat. No. 11754050), following the manufacturer’s instructions. The expression levels of the selected lncRNAs were quantified by real-time PCR using TaqMan^®^ gene expression assays (Thermo Fisher Scientific) in the StepOnePlus™ system (Thermo Fisher Scientific). *GAPDH* was used as the housekeeping gene (Appendix A), and the relative expression of lncRNAs and mRNA was calculated using the 2^−ΔΔCt^ method. Two independent experiments were performed.

### 2.12. lncRNA Knockdown

We studied the effects of knockdown of lncRNA *AGAP2-AS1* and *LINC01133* on the sensitivity of HCC827 and H1975 cell lines to TKIs. Briefly, cells were seeded in 48-well plates (1 × 10^5^ cells/well) and cultured overnight to allow attachment. After washing with RPMI without FBS, cells were cultured in serum-free medium for 4 h. Knockdown was performed with a TriFECTa RNAi Kit in OptiMEM medium (Thermo Fisher, Waltham, MA, USA, Cat. No. 31985) according to the provider’s instructions. The cells were incubated with 3 μL of Lipofectamine 3000 (Invitrogen, USA, Cat. No. L3000-015), and *AGAP2-AS1*, *LINC01133* DsiRNAs were added at a final concentration of 10 nM for 24 h, at which the maximum knockdown was detected. For each knockdown assay, we included a mock control (lipofectamine alone) and a DsiRNA for an unrelated gene (*HPRT1*). The DsiRNA sequences used for knockdown are provided in Appendix A. RT–qPCR was used to determine knockdown efficiency. Cell viability was evaluated after the exposure of transfected cells and controls to TKIs for 48 h using the MTT assay, as described above.

### 2.13. CD74 Localization in DTP Cells

The localization of CD74 in the DTP cells was determined using indirect immunofluorescence staining. Cells were cultured on four-chamber slides and treated as described above. After treatment, the cells were fixed with ethanol, washed, and treated for 30 min with a blocking solution to avoid nonspecific binding. The slides were incubated with anti-CD74 antibody (Cell Signaling Tech, MA, USA, Cat. No. 77274) at 1:100 dilution overnight in a humidified atmosphere at 4 °C. After washing, slides were incubated with Alexa Fluor 488-conjugated secondary antibody (1:500; Invitrogen, IL, USA, Cat. No. A2731) at 37 °C for 90 min. Raji cells were used as a positive control for CD74 staining. Finally, the cells were incubated with DAPI (1:150; Thermo Scientific, USA; Cat. No. 62248) for nuclear staining for 15 min. The slides were mounted with Vectashield (Vector Laboratories, Newark, CA, USA, Cat. No. H-1000). Micrographs were acquired using an EVOS FL microscope (Thermo Fisher Scientific, MA, USA). Two independent experiments were performed.

### 2.14. Tissue Samples of Lung Adenocarcinoma 

Specimens were obtained from untreated patients with advanced-stage primary lung adenocarcinoma (stage IIIb or IV). Tissue blocks from patients with lung adenocarcinoma were collected from the repository of residual biological material from the archives of the Pathology Department at the Instituto Nacional de Enfermedades Respiratorias Ismael Cosio Villegas. The biopsies selected for this study included those with a high proportion of tumor cells (70–90%). Two independent pathologists verified the diagnosis of lung adenocarcinoma according to the 2021 WHO Classification of Lung Tumors [37]. 

### 2.15. Validation of lncRNA in Lung Adenocarcinoma Biopsies

Four paraffin-embedded tissue biopsies from primary untreated LUAD patients were used for RNA extraction. Serial sections of 10 µm were processed using a PureLinkTM FFPE RNA isolation kit (Invitrogen, CA, USA), following the manufacturer’s instructions. The RNA concentration and purity were analyzed using a NanoDrop spectrophotometer (Thermo Fisher Scientific). Purified RNA was reverse-transcribed into cDNA using SuperScript™ VI-LO™ (Thermo Fisher Scientific, Cat. No. 11754050). The expression levels of the selected lncRNAs were quantified by real-time PCR using TaqMan^®^ gene expression assays (Thermo Fisher Scientific; Appendix A), as described in Section 2.11.

### 2.16. Immunohistochemical Staining of CD74

For immunostaining, slices were obtained from the collected biopsies of 15 untreated patients with *EGFR*-mutated lung adenocarcinomas. Tissue slices were deparaffinized at 60 °C in an oven for 20 min, followed by rehydration. Heat-induced epitope retrieval was carried out using 0.01 M citrate buffer (pH 6.0) in an NxGen decloaking chamber (BioCare Medical, Pacheco, CA, USA) at 110 °C and 6 psi for 20 min. All the slides were treated with 3% (*v*/*v*) H_2_O_2_ in methanol for 25 min to block endogenous peroxidase activity. After washing with PBS, slices were incubated with 2% serum-containing PBS for 30 min to block nonspecific binding. Immunostaining was performed using the anti-CD74 human monoclonal antibody (Cell Signaling, MA, USA, Cat. No. 77274) at a 1:100 dilution. The slides were incubated overnight at 4 °C in a humidified chamber, and the next day, the slides were washed twice with 0.1% *v*/*v* Tween in PBS and twice with PBS. The tissue sections were then incubated with a biotin-labeled anti-rabbit antibody (Thermo Fisher, CA, USA, Cat. No. 65-6140) at a 1:500 dilution in a humidified chamber at 32 °C for 2 h, washed, and incubated the VECTASTAIN Elite ABC-HRP kit at 1:300 for 30 min (Vector Laboratories, CA, USA, Cat. No. PK-6100). The color was developed using H_2_O_2_ as substrate and diaminobenzidine as chromogen. Slides were mounted with Entellan, and microscopy images were captured using a DFC425 C color camera (Leica Microsystems Inc., Wetzlar, Germany) coupled with a Leica DMLB light microscope and Leica Application Systems v. 3.6.0 software (Leica Microsystems Inc.).

### 2.17. Statistical Analysis

The results obtained under the experimental conditions are presented as mean ± standard deviation (SD). For comparison, the experimental and control groups were analyzed using the Student’s *t*-test. For multiple comparisons, one-way analysis of variance (ANOVA) was performed using Prism 10 (GraphPad Software, La Jolla, CA, USA). Statistical significance was set at *p* < 0.05. 

### 2.18. Data Availability Statement

The accession numbers of RNA-seq raw reads, raw count arrays, and target sequencing raw reads, as well as data analyses that were performed in R language using open-source packages, are available at the following link: https://github.com/maperezm/TKIS_Intrinsic_Resistance (accessed on 13 December 2023).

## 3. Results

### 3.1. Effects of TKIs on EGFR-Mutated Lung Adenocarcinoma Cell Lines

Microscopic observation of the cultures of HCC827 and HCC4006 cell lines revealed that the tested concentrations of erlotinib caused mainly a cytostatic effect (Figure 1A,D), observed as a gradual reduction in cell density after 48 h. Dose–response curves are shown in Appendix A. The highest concentration reduced cell density by approximately 50% compared to that of the control culture. A total of 100 nM (low) and 200 nM (high) concentrations of erlotinib were selected for further analysis since we observed cell-cycle arrest and increased proportion of cells in the G0/G1 phase, with a concomitant reduction in the S and G2 phases. In addition, the highest concentration of erlotinib induced a minimal proportion (10–20%) of apoptotic and necrotic cells (Appendix A), detected as a sub-G0 peak (Figure 1B–F). Our results suggest that S-phase and G2 cells are more susceptible to TKI-induced cell death (Figure 1C,F). Several reports have indicated that the H1975 cell line, which harbors the T750M mutation, is not affected by erlotinib. We exposed this cell line to erlotinib as a control of resistance. We confirmed that, under our experimental conditions, these cells were not susceptible to this drug, as the wide range of concentrations tested did not affect cell viability nor the proportion of cells in different phases of the cell cycle (Figure 1G–I). Therefore, erlotinib in the HCC827 and HCC4006 cell lines mainly caused a cytostatic effect (Appendix A).

It has been recently suggested that, regardless of the type of *EGFR* mutation detected in the tumor cells, LUAD patients should undergo osimertinib treatment [9,33]. Based on this proposal, we studied the effects of osimertinib in the cell lines employed. Compared to the effect induced by erlotinib, osimertinib induced a cytotoxic effect in the HCC827 and HCC4006 cell lines (Figure 2A,D). The highest osimertinib concentration (200 nM) induced approximately 60% cytotoxicity in both cell lines (Figure 2A–F and Appendix A). Surprisingly, in the H1975 cell line, which has been reported to be highly susceptible to osimertinib, this drug only altered the cell cycle by arresting cells in the G0/G1 phase without inducing cell death (Figure 2H,I).

Collectively, in the HCC827 and HCC4006 cell lines, erlotinib caused alterations in the cell-cycle, mainly blocking the transition from the G0/G1 phase to the S phase, whereas osimertinib induced a cytotoxic effect. The H1975 cell line, which harbors an erlotinib-resistance mutation, was not affected by this drug, whereas osimertinib only induced arrest in the G0/G1 phase. It should be noted that, despite the different sensitivities between the cell lines tested, residual cells were detected at the end of exposure in both treatments. The intrinsic resistance of these residual cells may be associated with the display of a drug-tolerant persister (DTP)-state. The subsequent experiments were performed to identify the characteristics that allow them to survive after treatment.

### 3.2. Osimertinib Induces β-Galactosidase Expression in Residual Cells

We found that cell-cycle arrest was an important characteristic of residual cells after TKI exposure. It has been reported that DTP cells acquire a non-proliferative senescent-like phenotype to survive exposure to therapeutic drugs [38]. As osimertinib is the treatment of choice for lung cancer harboring *EGFR* mutations [39], we investigated the expression of β-galactosidase, characteristic of the senescent-like phenotype, in the residual cells after exposure to osimertinib. Osimertinib-residual cells were cultured in fresh media for an additional five days and labeled for β-galactosidase expression. We found that more than 70% of the residual cells were β-galactosidase (β-gal)-positive (Figure 2 and Appendix A), suggesting that these cells acquired a senescent-like phenotype. In summary, cell-cycle arrest and β-gal expression on viable residual cells suggest that they may acquire a DTP state.

### 3.3. NGS-Based Mutation Analysis in Lung Adenocarcinoma Cell Lines

To exclude the possibility that driver mutations, distinct from *EGFR* mutations, could be associated with the biological response detected in the cell lines studied, we analyzed the cell lines employed for the mutation profiles of 15 solid cancer-associated genes. The results confirmed the presence of the corresponding *TP53*- and *EGFR*-activating mutations in the HCC827, HCC4006, and H1975 cell lines, as reported by the ATCC. As no additional mutations of clinical relevance were detected, we assumed that the biological behavior detected in these cell lines did not depend on any additional genetic alterations.

### 3.4. Transcriptomic Sequencing of Drug-Tolerant Persister Cells

To analyze the transcriptional changes associated with TKI resistance, we performed total RNA sequencing of the DTP and control cells. In total, 40 million reads per sample were generated, and read quality was analyzed using FastQC. High-quality reads were aligned, mapped, and estimated for abundance to ensure reproducibility between the biological replicates. Principal component analysis of the normalized counts revealed that the treatment was the main factor contributing to the variability of the dataset. We used edgeR to identify differentially expressed genes (DEGs). Significant changes in gene expression in erlotinib- and osimertinib-induced DTP cells were based on a |log2 fold change| ≥ 1 and an adjusted *p*-value < 0.05. The volcano plots for each cell line exposed to TKIs are shown in Figure 3A,B. 

We used Venn diagrams to identify the overexpressed genes shared among the DTP cells from the cell lines tested for each treatment (Figure 3C,D). In total, 71 and 87 genes were overexpressed in erlotinib- and osimertinib-induced DTP cells, respectively. Interestingly, we found that some of the overexpressed genes corresponded to lncRNAs. In total, 8 and 13 overexpressed lncRNAs were found in erlotinib- and osimertinib-induced DTP cells, respectively (Appendix A). Overexpression of these lncRNAs may account for an important mechanism of early transcriptomic reprogramming in multiple pathways that promote intrinsic resistance. Functional annotation of these commonly overexpressed RNAs using the Gene Ontology (GO) biological process database revealed that these lncRNAs are related to pathways involved in an extracellular matrix organization in erlotinib-induced DTP cells or adhesion-related pathways in osimertinib-induced DTP cells (Figure 3E,F). Thus, the overexpression of these lncRNAs could be related to epithelial–mesenchymal transition (EMT), which promotes intrinsic resistance. 

No statistically significant association with resistance-related pathways was found for the underexpressed genes of erlotinib-induced DTP cells. In contrast, GO analysis revealed that underexpressed genes of osimertinib-induced DTP cells are associated with pathways related to cell-cycle regulation (Appendix A). This finding is consistent with our data from cell-cycle assays where G0/G1 arrest was found on residual cells.

### 3.5. Survival Analysis

To explore the clinical relevance of the overexpressed lncRNAs, Kaplan–Meier (K–M) curves were generated from the LUAD project of TCGA. High expression of the erlotinib-DTP-associated lncRNAs *AGAP2-AS1* and *NKILA* was associated with poor OS. Similarly, high expression of the osimertinib-DTP-associated lncRNAs *CERS6-AS1*, *LINCO1133*, and the protein-coding gene *CD74* correlated with poor OS (Figure 4). The overexpression of these clinically relevant lncRNA and *CD74* was further validated in the DTP cells.

### 3.6. Validation of Clinically Relevant DTP-Associated lncRNAs and CD74

The expression levels of clinically relevant DTP-associated lncRNAs identified through RNA-seq were validated by RT–qPCR. The results confirmed the overexpression of *AGAP2-AS1* and *NKILA* in erlotinib-induced DTP cells, consistent with our RNA-seq results. Overexpression of *AGAP2-AS1* was higher in HCC827 cells than in HCC4006 cells, which may account for the different sensitivities to erlotinib observed in these cell lines (Figure 5A). However, this proposal requires more robust in vitro and in vivo research. 

In osimertinib-induced DTP cells, the expression levels of *CERS6-AS1* and *LINC01133* lncRNAs were consistent with the transcriptome sequencing results. Interestingly, the HCC827 cell line, which showed the highest osimertinib sensitivity among the cell lines, displayed the lowest overexpression of lncRNA *LINC01133*. Additionally, we found that the expression of the protein-coding gene *CD74* was notably increased in osimertinib-induced DTP cells from HCC4006 and H1975 cell lines (Figure 5B). These data were concordant with previous findings by Kashima et al., who used a single-cell assay for Transposase-Accessible Chromatin with sequencing (scATAC-seq analysis) and found that DTP cells from the H1975 cell line overexpressed CD74 after exposure to this TKI [40].

### 3.7. Knockdown of lncRNAs and CD74 Associated with the DTP State

To verify the involvement of the identified lncRNAs in the intrinsic resistance of LUAD cell lines to TKIs, we employed DsiRNA-mediated knockdown of these lncRNAs and *CD74*. Specifically, *AGAP-AS1*, which is overexpressed by the erlotinib-induced DTP cells, was knocked down in the HCC827 cell line, while *LINC01133* and *CD74*, which are overexpressed by the osimertinib-induced DTP cells, were knocked down in the H1975 cell line. Knockdown efficiency was assessed by RT–qPCR (Figure 5C). After transfection, the cells were exposed to the respective TKIs, and cell viability was assessed. Knockdown of the targeted lncRNAs and *CD74* significantly diminished cell viability in response to a single exposure to TKIs in comparison to the control conditions (Figure 5D).

These results substantiate the proposition that the overexpression of the proposed lncRNAs and *CD74* is linked to intrinsic resistance to TKIs. Furthermore, the increased sensitivity of the cells after knockdown suggests that these lncRNAs and *CD74* could be used as viable targets to overcome resistance to these therapeutic approaches.

### 3.8. Tissue Expression of Clinically Relevant lncRNAs Associated with the DTP State

To further explore the relevance of the DTP-associated lncRNAs, we analyzed their expression in lung adenocarcinoma biopsies using RT–qPCR. We assessed the expression levels of the clinically relevant lncRNAs associated with TKI-induced DTP cells in four formalin-fixed paraffin-embedded (FFPE) biopsies from nontreated LUAD patients. The osimertinib-induced DTP-associated lncRNAs in tissue samples were barely detectable compared to the erlotinib-induced DTP-associated lncRNAs, suggesting a lower amount of osimertinib-induced DTP cells in nontreated patients. This observation is in line with clinical trends, as erlotinib resistance is more frequent than osimertinib resistance in lung adenocarcinoma [41]. As we barely detected osimertinib-induced DTP-associated lncRNAs in biopsies, and our RNA-seq analysis in cell lines showed that the protein-coding gene *CD74* is overexpressed by the osimertinib-induced DTP cells, we also analyzed the expression of this protein-coding gene. The expression of *CD74* was elevated in tumor biopsies, similar to that observed in cell lines. This observation suggests that *CD74* overexpression may act as an additional mechanism in the cellular response of osimertinib-induced DTP cells, which may involve an intricate interplay between lncRNAs and protein-coding genes (Figure 6).

### 3.9. Cellular Localization of CD74 in Cell Lines

To evaluate the cellular localization of CD74 in the cell lines studied, we performed immunofluorescence staining for CD74 in the HCC4006 and H1975 cell lines. We found that CD74 is mainly located in the cytoplasm. CD74 staining increased in osimertinib-induced DTPs in accordance with the RT–qPCR results. Interestingly, heterogeneous staining of CD74 was observed within the same cell line, suggesting different expression levels among the tumor cell populations, which could be related to the different responses to osimertinib in the same cell line (Figure 7A).

### 3.10. CD74 Staining in lung Adenocarcinoma Biopsies

To assess whether CD74 is expressed in tumor tissues before treatment, we conducted immunohistochemical staining for CD74 in LUAD biopsies obtained from untreated patients. IHC staining revealed overexpression of CD74 in tumor cells compared to normal ciliated epithelial cells (Figure 7B). This observation suggests that CD74 overexpression is associated with the development of lung adenocarcinoma. Intriguingly, we observed that the tumor niche contains CD74 negative tumor cells and a lower proportion of CD74 positive cells (Figure 7C). Therefore, intrinsic resistance to osimertinib may depend on the presence of these CD74-expressing tumor cells. In other cases, CD74-positive cells with different staining intensities were observed, whereas tumor cells showing an invasive morphology were strongly positive (Figure 7D,E). However, a robust study using a larger number of biopsies and correlating their impact on patient survival is required to analyze the importance of CD74 protein expression.

In summary, our results suggest that, within the tumor, there are cells that are susceptible to TKIs, as well as tumor cells that overexpress molecules—such as CD74 and the DTP-associated lncRNAs—that allow them to resist the initial exposure to these therapeutic drugs, which is linked to intrinsic resistance. Eventually, these tumor cells give rise to new tumor clones that exhibit resistance to TKIs, promoting tumor relapse.

## 4. Discussion

Advances in the development of EGFR inhibitors (TKIs) with increased efficacy have encouraged the use of these drugs as first-line therapy for treating NSCLC harboring activating *EGFR* mutations. However, treatment failure frequently occurs due to the acquisition of molecular events that lead to resistance in tumor cells. Despite the application of new-generation TKIs, resistance to treatment has developed over time, leading to tumor relapse [42]. Numerous studies have explored the mechanisms of acquired resistance, usually analyzing this phenomenon after multiple rounds of exposure to therapeutic drugs [43,44,45]. Intrinsic resistance, denoting the inherent or natural resistance of cancer cells to specific treatments or drugs, has recently become a subject of exploration [46]. Initial investigations into this mechanism have been conducted in diverse lung cancer cell lines, with additional insights gleaned from information available in public databases [47,48]. In this context, the exploration of lncRNAs has emerged as a focal point of research. However, few studies have focused on the role of lncRNAs in intrinsic resistance to TKIs in lung cancer [46]. The present study sought to identify the transcriptional changes associated with DTP cells in lung adenocarcinoma cell lines induced after a single exposure to erlotinib or osimertinib.

The HCC827, HCC4006, and H1975 lung adenocarcinoma cell lines showed distinct sensitivities to a single exposure to therapeutic drugs. However, in the three *EGFR*-mutated lung adenocarcinoma cell lines, TKIs mainly inhibited cell proliferation, arresting cells in the G0/G1 phase of the cell-cycle. Consistent with our results, it has been reported that DTP cells display a reversible phenotype characterized by decreased cell proliferation [37]. Moreover, the acquisition of a non-proliferative senescent-like phenotype has been proposed as a mechanism of resistance of DTP cells to therapeutic drugs [26]. To explore whether the TKI-induced DTP cells acquired this phenotype after a single drug exposure, we assessed the enzyme β-galactosidase activity, which has been reported in drug resistance-associated senescence [49]. β-Galactosidase-positive DTP cells were consistently observed under experimental conditions, even for several days after removal of the drug. Since we observed that most cells produce this enzyme after drug exposure, our findings suggest that the residual cells display a DTP state.

To identify the transcriptional changes that led to the development of the DTP state in the cell lines, we analyzed and compared the transcriptomes of residual and control cells using RNA sequencing. To identify widespread mechanisms that induce intrinsic resistance, we evaluated the lncRNAs overexpressed in common by the erlotinib- or osimertinib-induced DTP cells. Interestingly, lncRNAs were found among the DEGs in DTP cells, and RT–qPCR validated their deregulation. Overexpression of these lncRNAs may be associated with early transcriptome reprogramming in DTP cells, which promotes intrinsic resistance. For instance, our results indicate that the HCC827 cell line is more resistant to erlotinib-induced cell death than HCC4006. Consistently, erlotinib-induced DTP cells from the HCC827 cell line displayed higher overexpression of lncRNA *AGAP2-AS1* than HCC4006 erlotinib-induced DTP cells. In contrast, the HCC827 cell line showed high sensitivity to osimertinib, and osimertinib-induced DTP cells from this cell line displayed low overexpression of the lncRNA *LINC01133*, in comparison to the H1975 osimertinib-induced DTP cells. In this regard, the knockdown of *AGAP2-AS1* in the HCC827 cell line and *LINC01133* in the H1975 cell line promoted the sensitivity of these cell lines to their respective TKIs. These results suggest that overexpression of these lncRNAs may be related to the intrinsic resistance to TKIs. The high expression of *AGAP2-AS1* and *LINC01133* was associated with significantly lower OS in the LUAD cohort of the TCGA, which supports the clinical relevance of these lncRNAs, as previously reported by several authors [50,51,52,53].

*AGAP2-AS1* inhibits apoptosis and promotes cell proliferation via the miR-1993a-3p/LOXL4 axis [54,55]. It has also been implicated in epithelial–mesenchymal transition (EMT) through the miR-468-3p/SRSF1 axis in colorectal cancer [56]. In LUAD, this lncRNA may also promote EMT as a mechanism of intrinsic resistance to erlotinib, as we found in our GO analysis. Accordingly, *AGAP2-AS1* has been proposed as a potential diagnostic and prognostic biomarker for NSCLC [50]. Dysregulation of *LINC01133* in NSCLC has been consistently observed in NSCLC cell lines and samples from both blood and tissues of patients with lung cancer. Zang et al. highlighted the significant overexpression of *LINC01133* expression in NSCLC tissues compared to healthy tissues. This up-regulation overexpression was also evident in the NSCLC cell lines A549, PC-9, and H1975, as opposed to the human bronchial epithelial cell line HBEC [52]. These findings strongly indicated that *LINC01133* functions as an oncogenic molecule in NSCLC. In addition, Zhang et al. [53] provided insights into the clinical implications of elevated *LINC01133* expression in NSCLC. Higher expression levels of *LINC01133* were positively correlated with several aggressive tumor characteristics, including larger tumor size, lymph node metastasis, and increased cell proliferation, migration, and invasion. Elevated *LINC01133* expression is associated with a higher tumor grade. Importantly, a study linked higher *LINC01133* expression to diminished overall survival in patients with NSCLC [53]. These studies emphasize the clinical relevance of *LINC01133* in NSCLC pathogenesis and its potential use as a prognostic marker. According to our results, we propose that *LINC01133* also be involved in intrinsic resistance to osimertinib.

To scrutinize the presence of the TKI-induced DTP-associated lncRNAs, which could be indicative of intrinsic resistance in nontreated LUAD patients, we explored their expression in tumor biopsies. Using RT–qPCR, we evaluated the expression levels of these lncRNAs. We found that erlotinib-induced DTP-associated lncRNAs *MEG3* and *AGAP2-AS1* could be detected in tumor biopsies from nontreated patients, suggesting that these lncRNAs may be overexpressed throughout tumor development and that patients displaying overexpression of these lncRNAs could develop resistance to erlotinib. Interestingly, osimertinib-induced DTP-associated lncRNAs *LINC01133* and *CERS6-AS1* could not be detected in the tumor biopsies. This observation is in line with clinical reports indicating that resistance to osimertinib is less frequent than resistance to erlotinib, supporting the use of osimertinib as a better therapeutic choice. It is possible that the tumor cells that express these osimertinib-induced DTP-associated lncRNAs are present in a low proportion prior to treatment and that the expression of these lncRNAs increases after the first round of treatment. 

Since we did not observe detectable levels of osimertinib-induced DTP-associated lncRNAs in tumor biopsies from untreated patients, we decided to investigate the expression of a protein-coding gene that could potentially contribute to intrinsic resistance to osimertinib. Our transcriptome analyses revealed *CD74* as a protein-coding gene overexpressed by the osimertinib-induced DTP cells. Kashima et al., using single-cell RNA analyses, identified *CD74* in the H1975 cell line to be associated with resistance to osimertinib [40]. Under our experimental conditions, the knockdown of CD74 caused a higher sensitivity to osimertinib in the H1975 cell line, suggesting an important involvement in the intrinsic resistance to this TKI. 

*CD74* is a protein-coding gene that encodes an invariant chain associated with HLA class II. This molecule, synthesized in the endoplasmic reticulum, is transported to the membranes of antigen-presenting cells (APCs). CD74 is a receptor for macrophage migration inhibitory factor (MIF) secreted by phagocytic cells [57]. In addition, it has been reported that CD74 is expressed mainly by the malignant cells of NSCLC and other cancers. Several CD74 in-frame fusion proteins have been reported, and this field is currently under intense investigation [58]. In the present study, RT–qPCR analysis confirmed *CD74* expression in osimertinib-induced DTP cells from cell lines and lung adenocarcinoma biopsies. It is worth noting that, in the HCC827 cell line, which exhibited high sensitivity to osimertinib, the expression of *CD74* was the lowest compared to those for the cell lines HCC4006 and H1975, which displayed higher resistance to osimertinib. 

To further investigate the role of CD74 in osimertinib resistance, we analyzed the expression of CD74 in the cell lines via immunofluorescence. CD74 was distributed in the cytoplasm and membrane, and its expression was higher in the osimertinib-induced DTP cells than in the control. Additionally, in tumor biopsies, we found that tumor cells displayed strong CD74 staining, while normal ciliated epithelial cells showed no staining, which suggests that expression of this molecule in epithelial cells is related to tumor development or progression. Interestingly, a heterogeneous staining pattern among the tumor cells was observed in the tumor mass, implying that some tumor cells expressed higher levels of CD74. Tumor cells that express higher levels of CD74 would survive after the initial rounds of osimertinib treatment and would then proliferate to promote tumor relapse and resistance to this therapeutic option. These results are in line with those reported by Kashima et al. [40] and strengthen the proposal of CD74 as a therapeutic target to overcome osimertinib resistance. Whether the role of CD74 in intrinsic resistance to osimertinib is related to the multiple CD74 fusion proteins reported in lung cancer remains to be investigated.

## 5. Conclusions

Long non-coding RNAs (lncRNAs) have been implicated in drug resistance, potentially participating since the onset of treatment. After a single exposure to TKIs, the transcriptomic analysis revealed in the TKI-induced-DTP cells overexpression of some lncRNAs and the protein-coding gene *CD74* The clinical relevance of the *AGAP2-AS1*, *LINC001133*, and *CD74* was validated in a cohort of patients from the TCGA database. Also, the overexpression and participation of these genes were validated in DTP cells using RT–qPCR, and through knockdown experiments. In addition, these genes were detected in a small cohort of biopsies from nontreated patients with LUAD. Further basic and clinical practice studies are required to deepen the knowledge of their role in the intrinsic resistance of lung adenocarcinomas. However, this research provides opportunities to study the involvement of lncRNAs in the intricate genetic and epigenetic mechanisms associated with intrinsic resistance. We expect that these findings will help overcome the TKI resistance and thus improve the survival of LUAD patients.

## Figures and Tables

**Figure 1 cancers-16-02490-f001:**
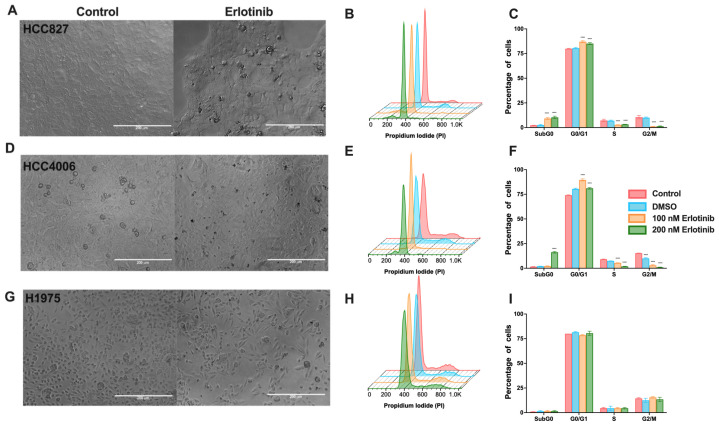
Morphological changes and proportion of cells in the cell-cycle phases in *EGFR*-mutated lung adenocarcinoma cell lines after a single erlotinib exposure. Cellular density changes induced by 200 nM erlotinib (**A**,**D**,**G**). Magnification 20×. Histograms showing the cell-cycle distribution profile and graphs of the percentage of cells in the cell-cycle phases induced by erlotinib in the HCC827 (**B**,**C**), HCC4006 (**E**,**F**), and H1975 (**H**,**I**) cell lines. Data are shown as the mean ± SD. **** *p* < 0.0001.

**Figure 2 cancers-16-02490-f002:**
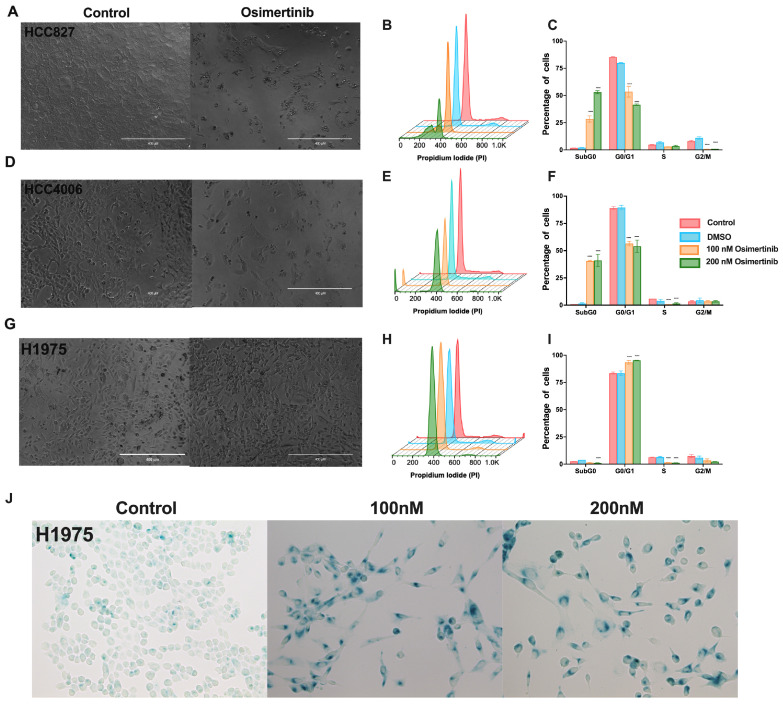
Effect of osimertinib exposure on *EGFR*-mutated lung adenocarcinoma cell lines. Morphological changes induced by single 200 nM osimertinib exposure (**A**,**D**,**G**). Magnification 20×. Histograms show the profile of cell distribution and graphs of the percentage of cells in the cell-cycle phases and the sub-G0 peak induced by osimertinib in the HCC827 (**B**,**C**), HCC4006 (**E**,**F**), and H1975 (**H**,**I**) cell lines. Data are shown as mean ± SD. **** *p* < 0.0001. (**J**). After a single 48 h exposure to osimertinib, residual cells from the H1975 cell line were cultured in fresh complete media for five days, and the expression of beta-galactosidase was detected. Micrographs show β-gal-positive cells. Magnification 40×.

**Figure 3 cancers-16-02490-f003:**
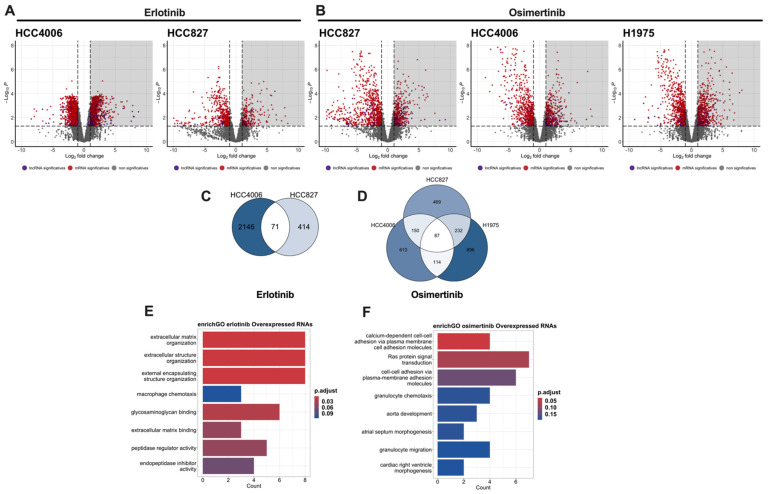
Differentially expressed genes (DEGs) in DTP cells. Volcano plots showing DEGs in DTPs relative to untreated cells. Gray shading indicates overexpressed transcripts (**A**,**B**). Venn diagrams of the overexpressed genes shared among DTP cells (**C**,**D**). Functional annotation of the overexpressed lncRNAs was performed using the Gene Ontology (GO) biological process database (**E**,**F**).

**Figure 4 cancers-16-02490-f004:**
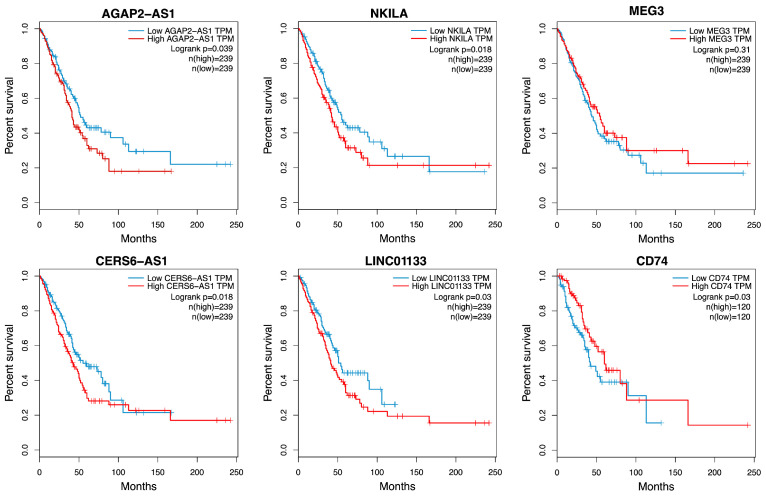
Clinical relevance of lncRNAs and *CD74* associated with intrinsic resistance. K–M survival analysis of the LUAD patient cohort from The Cancer Genome Atlas. High expression is shown in red, and low expression is shown in blue. Log-rank *p* values are indicated.

**Figure 5 cancers-16-02490-f005:**
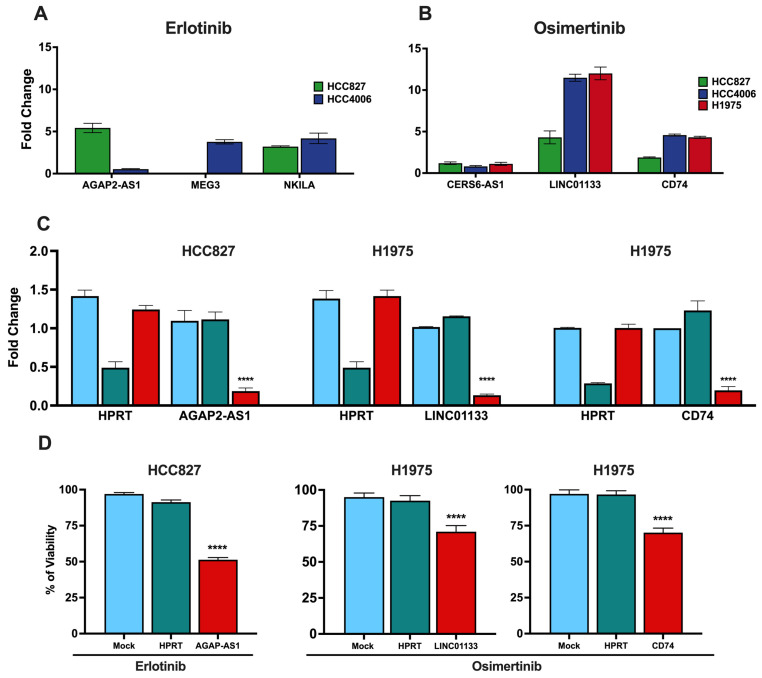
Gene expression of clinically relevant lncRNAs detected in DTP cells. Expression of the lncRNAs detected in the erlotinib- and osimertinib-induced DTP cells. Fold changes relative to the control are shown (**A**,**B**). DsiRNA-mediated knockdown reduced the expression of lncRNAs *AGAP2-AS1* in the HCC827 cell line and *LINC01133* and *CD74* in the H1975 cell line (red). Mock (light blue) and DsiRNA against *HPRT* (green) were used as controls (**C**). Knockdown of the DTP-associated lncRNAs reduced cell viability after a single exposure to TKIs (**D**). Data are shown as mean ± SD of three independent experiments. **** *p*-value < 0.05.

**Figure 6 cancers-16-02490-f006:**
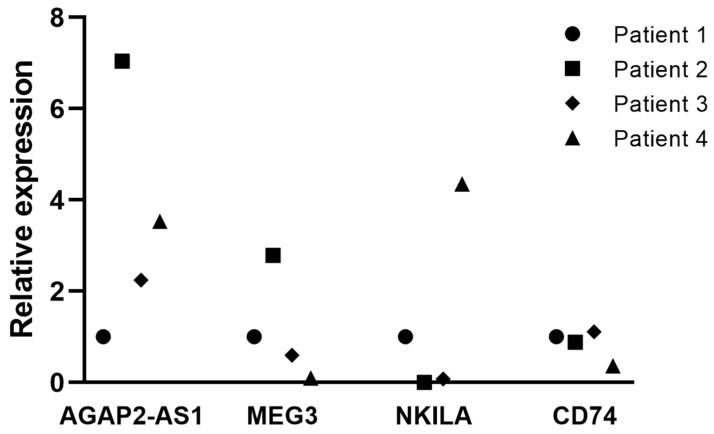
Expression of clinically relevant lncRNAs and CD74 in tumor biopsies. Expression of DTP-associated genes in four FFPE lung adenocarcinoma biopsies detected using RT–qPCR. Gene expression was normalized to GAPDH and compared to that of Patient 1.

**Figure 7 cancers-16-02490-f007:**
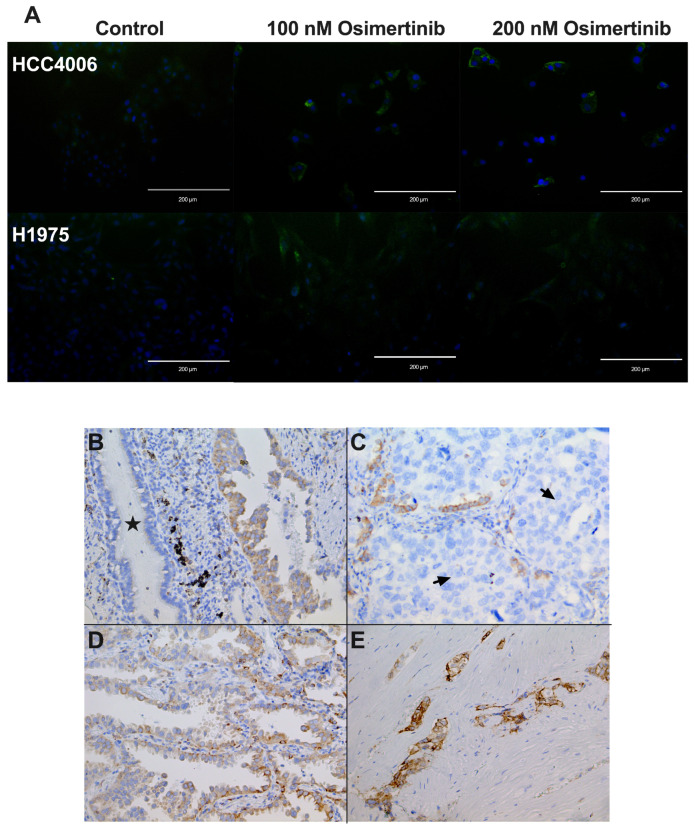
CD74 expression in lung adenocarcinoma. Immunofluorescence analysis of CD74 in osimertinib-induced DTP cells. CD74 mainly showed cytoplasmic staining, which increased in osimertinib-induced DTP cells. CD74 (green) and nuclear staining (blue). Magnification 20× (**A**). Lung adenocarcinoma biopsies. Comparison of CD74 staining between normal ciliated epithelial cells (star) and malignant cells (**B**). CD74-negative (arrow) and CD74-positive tumor cells within the same niche (**C**). Tumor cells expressing varying levels of CD74 (**D**). High CD74 staining was observed in tumor cells with invasive morphology (**E**). Magnification 20×.

## Data Availability

The accession numbers of RNA-seq raw reads, raw count arrays, target sequencing raw reads, and data analyses performed in R language using open-source packages are available at the following link: https://github.com/maperezm/TKIS_Intrinsic_Resistance (15 March 2024).

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
