# Peer review of "Transcriptomic Analysis Reveals Early Alterations Associated with Intrinsic Resistance to Targeted Therapy in Lung Adenocarcinoma Cell Lines"

_cancers, 2024, doi:10.3390/cancers16132490_

Round 1

Reviewer 1 Report (Previous Reviewer 1)

Comments and Suggestions for Authors

Accept in present form

Author Response

Dear reviewer:
We appreciate the time spent in reviewing the manuscript.

Dolores Aguilar-Cazares

Reviewer 2 Report (Previous Reviewer 2)

Comments and Suggestions for Authors

The manuscript "Transcriptomic Analysis Reveals Early Alterations Associated with Intrinsic Resistance to Target Therapy in Lung Adenocarcinoma Cell Lines" presents a nuanced investigation into the mechanisms of intrinsic resistance within lung adenocarcinoma, emphasizing the pivotal role of long non-coding RNAs (lncRNAs) in the emergence of drug-tolerant persister (DTP) cells. This research enriches our current understanding of lung adenocarcinoma's resilience against tyrosine kinase inhibitors (TKIs) such as Erlotinib and Osimertinib, laying the groundwork for potential therapeutic advancements. However, despite its valuable contributions, the study has several areas that warrant critical appraisal and enhancement:

1.       The inhibitory curves of the inhibitors should be evaluated to determine their maximum inhibitory concentrations. The optimal concentration for each inhibitor should be selected based on these curves.

2.       The updated manuscript still does not address the significant phenotypic differences revealed by cell cycle analyses among the HCC827, HCC406, and H1975 cell lines, each with distinct EGFR statuses. These differences make it impractical to use the shared genes affected by Osimertinib treatment across these lines as a basis for investigating underlying mechanisms. Specifically, in the H1975 cell line, the treatment induces cell senescence rather than drug resistance, as indicated by the data. Moreover, the treatment's effects are not limited to gene overexpression; a substantial number of genes are downregulated, as shown in the volcano plot. Therefore, focusing exclusively on overexpressed genes overlooks crucial aspects of the treatment's impact, indicating that a more comprehensive analysis is necessary to accurately understand the effects.

3.       Section 3.4 lacks the explanations of the GO analyses.

4.       The manuscript lacks the western blots of cells with knocked down lncRNAs.

5.       There is no rationale for comparing related genes among patients who did not receive treatment with Erlotinib or Osimertinib.

Comments on the Quality of English Language

The quality of the English should be optimized, and the grammar should be thoroughly checked throughout the manuscript.

Author Response

Dear Reviewer:

In response to the valuable comments, we have substantially revised the manuscript. The major changes are shown in the red-colored manuscript.

  1. The inhibitory curves of the inhibitors should be evaluated to determine their maximum inhibitory concentrations. The optimal concentration of each inhibitor should be selected based on these curves.

We welcomed your comments. In this manuscript version, we have included the viability curves for each cell line treated with erlotinib and osimertinib. The inhibition curves and analysis of the maximum concentrations used are included in lines 210-211 and 352-353, in red text, and the analyses of the curves are shown in Fig S1 in Supplementary Material.

  1. The updated manuscript still does not address the significant phenotypic differences revealed by cell cycle analyses between the HCC827, HCC406 and H1975 cell lines, each with distinct EGFR states. These differences make it impractical to use the shared genes affected by osimertinib treatment in these lines as a basis for investigating the underlying mechanisms. Specifically, in the H1975 cell line, treatment induces cellular senescence rather than drug resistance, as the data indicate. Moreover, treatment effects are not limited to gene overexpression; a substantial number of genes are downregulated, as shown in the volcano diagram. Therefore, focusing exclusively on overexpressed genes misses crucial aspects of the impact of treatment, indicating that further analysis is needed to accurately understand the effects.

Thank you for your comment. We have updated the manuscript to clarify the observed differences between lung adenocarcinoma cell lines HCC827, HCC4006, and H1975 that harbor EGFR mutations. The HCC827 and HCC4006 cell lines harbor the most frequent EGFR mutation, an exon 19 deletion, which confers sensitivity to first-generation TKIs such as erlotinib. In contrast, the H1975 cell line has a double mutation (L858R/T790M), which confers resistance to erlotinib but sensitivity to the third-generation TKI osimertinib.

We recognize that osimertinib treatment induces variable phenotypic responses, including cellular senescence in the H1975 cell line. Reports indicate that DTP cells may acquire a senescence phenotype, allowing them to resist osimertinib exposure. Despite these phenotypic differences, in our search for markers and mechanisms of TKIs-induced early resistance in diverse tumor scenarios with different EGFR mutations, we focused on overexpressed lncRNAs. Our rationale is that overexpressed lncRNAs could be potential biomarkers of treatment response or regulators of mechanisms leading to stable resistance. Such biomarkers can be more easily identified in patients for treatment monitoring purposes than gene signatures.

Although we agree that downregulated genes may also be crucial in resistance mechanisms, our differential analysis did not identify statistically significant downregulated lncRNAs. We have revised the manuscript to highlight these points explicitly. We plan to conduct further studies to explore the role of downregulated genes in resistance mechanisms in future research.

3.- Explanations on GO analyses are missing in section 3.4.

Thank you for your comment. The requested explanation has been included in the revised version of the manuscript. This point has been resolved in lines 443-454.

4.- The manuscript lacks the western blots of cells with deleted lncRNAs.

Many thanks for your valuable comments. We appreciate your suggestion regarding including western blots for cells with deleted lncRNAs. However, as lncRNAs are RNA molecules, western blotting, a protein-specific technique, cannot be used to evaluate them. To validate the knockout of lncRNAs, we performed quantitative reverse-transcription PCR (RT-qPCR). This technique is well-suited for measuring RNA levels and allows us to confirm the successful knockout of target lncRNAs. We have included the qRT-PCR results in the manuscript to demonstrate the efficacy of the lncRNA silencing (Figure 5).

  1. There is no rationale for comparing related genes among patients who did not receive treatment with Erlotinib or Osimertinib.

Thank you for your comments. It has been proposed that specific molecules involved in intrinsic resistance mechanisms are present in tumor cells before exposure to treatment. Owing to tumor heterogeneity, when treatment is administered, tumor cells expressing these molecules may persist, evade treatment, and contribute to residual disease and tumor relapse. To investigate this, we evaluated the expression of lncRNAs and CD74 in biopsies obtained from untreated patients. This approach allowed us to determine whether these molecules were present as potential intrinsic resistance mechanisms before treatment initiation.

I want to indicate that native English-speaking editors from MDPI author services have reviewed the manuscript.

Best regards.

Sincerely,

Dolores Aguilar-Cazares

Round 2

Reviewer 2 Report (Previous Reviewer 2)

Comments and Suggestions for Authors

The authors addressed the comments previously provided.

Comments on the Quality of English Language

The language is good although there are some minor typos

This manuscript is a resubmission of an earlier submission. The following is a list of the peer review reports and author responses from that submission.

Round 1

Reviewer 1 Report

Comments and Suggestions for Authors

1. Title and Simple Summary:

  1. The title effectively summarizes the focus of the study. 
  2. In the simple summary, it's recommended to specify the lung adenocarcinoma cell lines used for the study (if not mentioned elsewhere) for clarity.

2. Abstract:

  1. The abstract provides a concise overview of the study, including objectives, methods, and key findings. It effectively highlights the significance of the research.
  2. "lncRNAs" should be expanded upon first mention as "long non-coding RNAs."
  3. Specify the abbreviation "DTP" upon first mention.
  4. In the last sentence, "OS" should be expanded upon first mention, as it typically stands for "overall survival" in oncology research.

3. Introduction:

  1. The introduction provides relevant background information on lung adenocarcinoma, targeted therapies, and the problem of drug resistance.
  2. The transition between discussing NSCLC and targeted therapies could be smoother.
  3. Consider providing more specific statistics or references for the reported rates of drug resistance to strengthen the argument.
  4. Define "DTP" upon first mention.
  5. The discussion of tumor heterogeneity and DTP cells is well-supported with references.
  6. Clarify the significance of the proposed study in addressing the gaps in understanding intrinsic resistance and the role of lncRNAs.

4. Results:

  1. Subsection 3.1: The presentation of the effects of TKIs on different cell lines is clear. The inclusion of figures enhances understanding.
  2. Ensure consistency in the presentation of figure references (e.g., "Figure 1A" vs. "Figures 2A and D").
  3. Subsection 3.2: The observation of β-galactosidase expression in residual DTP cells is significant. Provide more context on the implications of this finding
  4. Subsection 3.3: The mutation analysis subsection provides necessary validation of the cell lines used.
  5. Subsection 3.4: The description of transcriptomic sequencing is clear. Ensure consistency in the use of terminology (e.g., "DTP cells" vs. "residual cells").
  6. Subsection 3.5: The survival analysis is crucial for linking findings to clinical relevance. Provide more context on how these findings contribute to understanding patient outcomes.
  7. Subsection 3.6: The gene expression validation is essential for confirming transcriptomic findings. Ensure clarity in explaining the relevance of specific lncRNAs.
  8. Subsection 3.7: The knockdown experiments provide valuable insights into the functional role of identified lncRNAs. Clarify the implications of increased sensitivity after knockdown.
  9. Subsection 3.8: The analysis of tissue expression adds clinical relevance. Provide more context on how these findings translate to patient care.
  10. Subsection 3.9: The exploration of CD74 expression expands the scope of the study. Clarify the potential implications of CD74 overexpression in tumor development.
  11. Subsection 3.10: The examination of CD74 staining in biopsies strengthens the clinical relevance. Provide more context on how CD74 expression relates to treatment outcomes.

5. Discussion:

  1. The discussion effectively synthesizes the findings and places them in the context of existing literature.
  2. Consider providing a more detailed interpretation of the results and their implications for future research and clinical practice.
  3. Ensure clarity in discussing the role of lncRNAs and CD74 in intrinsic resistance, potential therapeutic implications, and areas for further investigation.

6. General Comments:

  1. The manuscript demonstrates a comprehensive investigation into the mechanisms of intrinsic resistance in lung adenocarcinoma.
  2. Ensure consistency in terminology, abbreviations, and formatting throughout the manuscript.
  3. Proofread the manuscript for grammatical and typographical errors.
  4. Consider adding subsection headings within sections to improve readability and organization.

Here are useful references for authors observation:

https://www.sciencedirect.com/science/article/pii/S0378517322005828

https://www.ncbi.nlm.nih.gov/pmc/articles/PMC6934956/

Comments on the Quality of English Language

Minor editing of English language required

Reviewer 2 Report

Comments and Suggestions for Authors

The manuscript "Transcriptomic analysis reveals early alterations associated with intrinsic resistance to target therapy in lung adenocarcinoma cell lines" presents a nuanced investigation into the mechanisms of intrinsic resistance within lung adenocarcinoma, spotlighting the pivotal role of long non-coding RNAs (lncRNAs) in the emergence of drug-tolerant persister (DTP) cells. This research enriches the current understanding of lung adenocarcinoma's resilience against tyrosine kinase inhibitors (TKIs) such as erlotinib and Osimertinib, laying the groundwork for potential therapeutic advancements. However, despite its valuable contributions, the study offers room for critical appraisal and enhancement in several areas:

1.      There are grammar mistakes and spelling errors throughout the manuscript.

2.      The plots and legends are hard to read throughout the manuscript.

3.      The cell cycle analyses reveal significant phenotypic differences among the HCC827, HCC406, and H1975 cell lines. Given these distinctions, it seems impractical to utilize the shared genes impacted by Osimertinib treatment across these three lines as a basis for investigating underlying mechanisms. This is particularly relevant as the treatment induces cell senescence in the H1975 line, rather than drug resistance, as evidenced by the data. Furthermore, the treatment's effect is not limited to gene overexpression; a considerable number of genes are downregulated, as illustrated in the volcano plot. Thus, focusing exclusively on overexpressed genes overlooks critical aspects of the treatment's impact, suggesting a more comprehensive analysis is warranted to accurately understand the effects.

4.      In the survival analyses concerning the selected genes, the authors aim to demonstrate a correlation between these genes and survival outcomes. However, it would be more effective to develop a gene signature comprising these genes to illustrate their collective association with survival. This approach, rather than individually detailing each gene's correlation, would provide a more comprehensive understanding of how these genes, as a group, impact survival rates.

5.      While the authors have identified overexpression of CD74 in both cell lines and tumor tissues, the presented data falls short of conclusively demonstrating a link between CD74 overexpression and drug resistance. Additional evidence and more in-depth analyses are needed to firmly establish this association and understand its implications for drug resistance mechanisms.

Comments on the Quality of English Language

1.      There are grammar mistakes and spelling errors throughout the manuscript.

2.      The plots and legends are hard to read throughout the manuscript.